# Polygenic score accuracy in ancient samples: Quantifying the effects of allelic turnover

**Maryn O. Carlson**[1]*, **Daniel P. Rice**[2], **Jeremy J. Berg**[1,2], **Matthias Steinrücken**[1,2,3]*

**1** Committee on Genetics, Genomics, & Systems Biology, University of Chicago, Chicago, Illinois, United States of America, **2** Department of Human Genetics, University of Chicago, Chicago, Illinois, United States of America, **3** Department of Ecology & Evolution, University of Chicago, Chicago, Illinois, United States of America

* mocarlson@uchicago.edu (MOC); steinrue@uchicago.edu (MS)

**Data Availability Statement:** All code to generate the results is available at https://github.com/marync/ancient_polygenic.

**Funding:** MOC received funding from a National Institute of Health training grant T32 GM07197.

## Abstract

Polygenic scores link the genotypes of ancient individuals to their phenotypes, which are often unobservable, offering a tantalizing opportunity to reconstruct complex trait evolution. In practice, however, interpretation of ancient polygenic scores is subject to numerous assumptions. For one, the genome-wide association (GWA) studies from which polygenic scores are derived, can only estimate effect sizes for loci segregating in contemporary populations. Therefore, a GWA study may not correctly identify all loci relevant to trait variation in the ancient population. In addition, the frequencies of trait-associated loci may have changed in the intervening years. Here, we devise a theoretical framework to quantify the effect of this allelic turnover on the statistical properties of polygenic scores as functions of population genetic dynamics, trait architecture, power to detect significant loci, and the age of the ancient sample. We model the allele frequencies of loci underlying trait variation using the Wright-Fisher diffusion, and employ the spectral representation of its transition density to find analytical expressions for several error metrics, including the expected sample correlation between the polygenic scores of ancient individuals and their true phenotypes, referred to as polygenic score accuracy. Our theory also applies to a two-population scenario and demonstrates that allelic turnover alone *may* explain a substantial percentage of the reduced accuracy observed in cross-population predictions, akin to those performed in human genetics. Finally, we use simulations to explore the effects of recent directional selection, a bias-inducing process, on the statistics of interest. We find that even in the presence of bias, weak selection induces minimal deviations from our neutral expectations for the decay of polygenic score accuracy. By quantifying the limitations of polygenic scores in an explicit evolutionary context, our work lays the foundation for the development of more sophisticated statistical procedures to analyze both temporally and geographically resolved polygenic scores.

## Author summary

The genomes of ancient organisms document, albeit imperfectly, the migrations, admixture events, and displacements that may have occurred in a given species' history.

The funders had no role in study design, data collection and analysis, decision to publish, or preparation of the manuscript.

**Competing interests:** The authors have declared that no competing interests exist.

Researchers also use these ancient genomes to learn whether genetic changes underlie the evolution of polygenic traits, like height and disease susceptibility, which are affected by many genetic variants of small effects. Such analyses rely on models, often garnered from large-scale genetic association studies, that predict the phenotypes of ancient individuals from their genotypes. Yet, theoretical and empirical research suggests that prediction accuracy depends on the relationship between the sample in which the model was built and the sample in which it is applied. In this vein, we quantify the effects of one fundamental limitation on prediction accuracy: the fact that allele frequencies may differ across time and geographic space. As a consequence, a given prediction model may not capture all of the genetic variation relevant to phenotypic variation in the focal, ancient sample.

## Introduction

Decay in linkage disequilibrium (LD) between tagging and causal sites, population stratification, variation in allele frequencies within and across populations, and environmental heterogeneity, among other factors, are all thought to negatively impact the prediction accuracy of polygenic scores (see e.g., [1–7], and more recently in humans, e.g., [8–13]). Many of these issues likely influence both within- *and* out-of-sample predictions, where out-of-sample may refer to an individual sampled from a distinct time or location relative to that of the GWA study. While empirical [12, 14] and simulation [1, 13, 15] or combined [16] studies have explored particular population genetic scenarios or experimental contexts, we still do not know the extent to which each of these factors compromises prediction accuracy in general.

In this work, we address an issue pertinent to out-of-sample prediction: that causal loci may have different allele frequencies in the GWA study and focal populations. Variants common in the GWA study may be rare in the focal population, and vice versa. We refer to this phenomenon as *allelic turnover*. Allelic turnover implies that effect estimates ported across space and time, or both, may not reflect all of the genetic variation relevant to phenotypic variation in an ancient or geographically distinct population. Allelic turnover further suggests that the statistical properties of ancient polygenic scores depend on when an ancient individual was sampled —a feature not currently accounted for in ancient DNA analyses. Similarly, statistical properties of geographically disparate polygenic scores depend on the divergence time between the GWA study and focal populations. An understanding of allelic turnover in these contexts may ultimately improve statistical analyses of temporally (e.g., [17–20]) and geographically resolved polygenic scores (e.g., [9, 10]), analyses which are increasingly commonplace.

We aim to quantify the effect of allelic turnover on the polygenic scores of such out-of-sample individuals when they are computed using effect estimates from a contemporary population. We expect that increases in ancient sampling time or divergence time will be associated with declines in polygenic score accuracy due exclusively to allelic turnover. The question is, by how much does accuracy decline? And, can allelic turnover alone explain the reduced accuracy of out-of-sample predictions observed in numerous human (e.g., [15, 16]), animal (e.g., [1, 2, 4]) and plant (e.g., [21, 22]) experiments and simulation studies. The answer is likely to depend on the particular population genetic, trait, and GWA study features of the system under study [3]. We attempt to capture some important aspects of this diversity in our modeling framework.

Here, we consider a standard implementation of the polygenic score $\hat{Y}$ which attributes non-zero effects to a particular set of loci, $\mathcal{S}$. An individual's polygenic score is a weighted sum of its genotype, where the weights are the estimated allelic effects. The loci in $\mathcal{S}$ and their

estimated effects are usually identified in large-scale GWA studies, often performed in regional biobanks with sample sizes in the tens to hundreds of thousands of individuals (e.g., the UK Biobank [23], BioBank Japan [24]). Frequently, the set $\mathcal{S}$ includes loci which are approximately independent and surpass some allele frequency and $p$-value thresholds. Though there are numerous ways to define a polygenic score (e.g., [25, 26] and see Section 4 in S1 Text), the "prune and threshold" method is commonly used and proves analytically tractable in our framework.

Previous quantitative genetic approaches, such as [27] and [16], largely ignore the underlying population genetic dynamics. For example, Wang et al. [16] estimate the reduction in polygenic score accuracy in a focal population relative to the GWA study population as a function of the fixed population-specific trait heritabilities, allele frequencies, and LD patterns, and the estimated per-locus effects. In contrast, we embed the ancient polygenic score in an explicit population genetic framework, allowing us to take into account changes in allele frequency as well as the statistical constraint imposed by a finite GWA study sample size. And, distinct from previous approaches to the evolutionary modeling of polygenic scores [28], we track the frequencies of *all loci* that potentially contribute to a trait—not just the loci included in the polygenic score (i.e., loci in $\mathcal{S}$).

Henceforth, we frame our study in terms of ancient polygenic scores. However, we formally demonstrate that our theoretical results apply to out-of-space polygenic scores, where the population divergence time multiplied by two is analogous to the ancient sampling time (see Fig 1 and Section 1 in S1 Text). The latter scenario can represent an ancient individual sampled from a population not directly ancestral to that of the GWA study as the two populations must have diverged at some point in the past. This scenario, to a first approximation, describes the population displacement events thought to be ubiquitous in the history of humans (e.g., [29]). However, human history is additionally characterized by numerous admixture events and population size changes (e.g., [29]) which are not yet captured within our modeling framework.

We use several statistics to characterize ancient polygenic score error in distinct population genetic and GWA study scenarios. Each statistic is indexed by the ancient sampling time $\tau$: the bias, $bias(\tau)$, mean-squared error, $mse(\tau)$, estimated additive genetic variance, $\hat{V}_A(\tau)$, and polygenic score accuracy, $\rho^2(\tau)$, which approximates the expectation of the squared sample correlation coefficient between the polygenic scores and phenotypes of an ancient sample. In addition, we can readily express these statistics as functions of the genetic divergence between the ancient and GWA study populations, as measured by the fixation index, $F_{ST}$ (Section 11 in S1 Text). We first derive general forms for these statistics that are agnostic to almost all of our modeling assumptions and which provide conceptual insights into the effects of allelic turnover. Next, we derive explicit, parameter-dependent expressions for each statistic when the trait is neutrally evolving in a population of constant size subject to recurrent mutation— which for small mutation rates approximates the infinite sites model. We take advantage of the spectral representation of the transition density function of the Wright-Fisher diffusion (*tdf*) to execute these computations [30–33]. We then find interpretable linear approximations for the initial rate of increase (or decrease) of the metrics under study. These approximations apply for the small ancient sampling times typical of ancient humans remains (e.g., see [18]).

Consistent with our expectations, $mse(\tau)$ increases and the estimated additive genetic variance $\hat{V}_A(\tau)$ decreases with increasing sampling age $\tau$. Despite the fact that $mse(\tau)$ and $\hat{V}_A(\tau)$ are measuring distinct quantities—and indeed have different functional forms—our linear approximations reveal that, under our assumptions, both statistics initially change at approximately the same rate. This rate is proportional to the product of the mutation rate and the

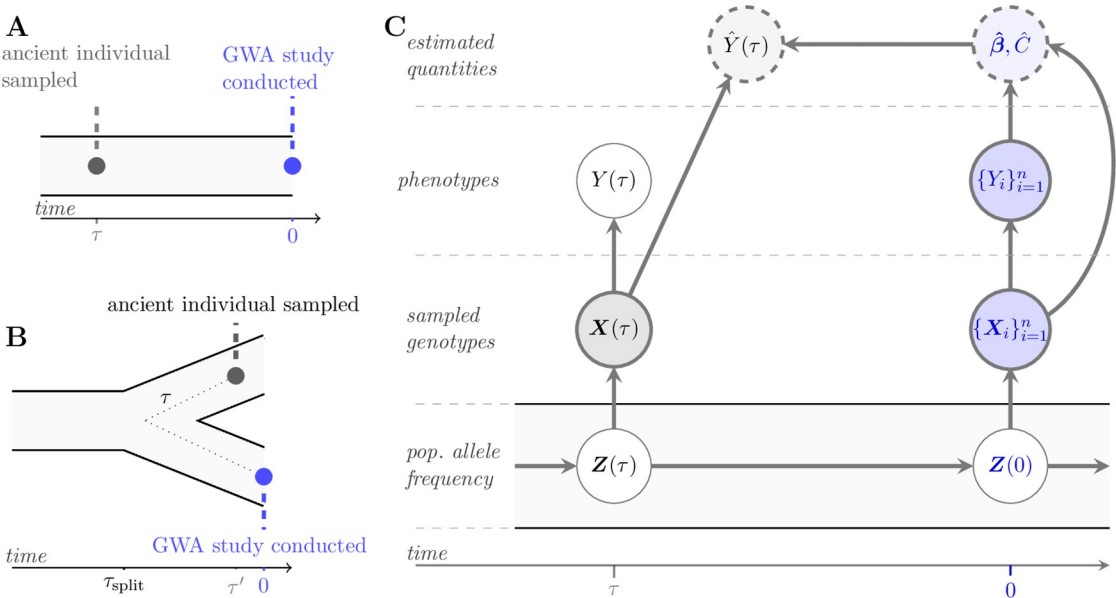

**Fig 1. A population genetic model for an ancient polygenic score.** Figures (A) and (B) portray the two demographic scenarios encompassed by our modeling framework. In (A), the ancient individual is sampled at an earlier time $\tau$ from the same population in which the GWA study is conducted. In (B), the ancient individual is sampled at an arbitrary time $\tau'$ from a population that split from the population in which the GWA study was conducted at some time $\tau_{\mathrm{split}}$ in the past. The dotted line schematically relates $\tau'$ to the ancient sampling time $\tau$ of (A), i.e., $\tau = 2\tau_{\mathrm{split}} - \tau'$. In (C), a graphical model relates the random variables explicit and implicit in the polygenic score $\hat{Y}(\tau)$ and phenotype $Y(\tau)$ of an ancient individual sampled $\tau$ generations in the past, as in (A). Darkly shaded and thickly bordered nodes are observed quantities. Unshaded and thinly bordered nodes are unobserved. Lightly shaded nodes bordered by dashed lines denote estimated quantities. Edges denote direct dependencies between connected nodes. For example, conditional on the ancient genotype $X(\tau)$, the polygenic score $\hat{Y}(\tau)$ is independent of the population allele frequencies $Z(\tau)$. Quantities in blue are associated with the present day only, and include the population allele frequencies $Z(0)$; the genotypes of the $n$ individuals in the GWA study, $\{X_i\}_{i=1}^n$ and their phenotypes, $\{Y_i\}_{i=1}^n$; and, the effects and intercept term estimated in the GWA study, $\hat{\beta}$ and $\hat{C}$, respectively.

power to detect trait-associated loci in the GWA study, which in turn, is influenced by both study size, the magnitude of the true per-locus effect, and the underlying distribution of the allele frequencies of causal loci.

Moreover, we show that polygenic score accuracy $\rho^2(\tau)$ is proportional to $\hat{V}_A(\tau)$, which, as stated, is sensitive to the GWA study and evolutionary parameters. Unlike $\hat{V}_A(\tau)$, $\rho^2(\tau)$ depends on the trait heritability $h^2$, with larger values of $h^2$ increasing its rate of decay. In contrast, for small mutation rates, relative accuracy, defined as the ratio of $\rho^2(\tau)$ to accuracy measured in a present-day sample $\rho^2(0)$, is insensitive to $h^2$, the true per-locus effect size, and the GWA study parameters, as long as the GWA study size $n$ exceeds some minimum threshold. We show that this result likely holds for an arbitrary distribution of effects. Importantly, accuracy and relative accuracy decay considerably over the short time spans characteristic of ancient human samples and geographically distinct human populations.

With equal probability of detecting positive versus negative effect alleles, and under neutrality, the bias of the polygenic score is zero for all ancient sampling times. In practice, both of these conditions are likely violated. For example, detection imbalances have been observed in case-control GWA studies [34], and many polygenic traits are likely under some form of selection [35, 36]. While unequal thresholds do not precisely capture the phenomena described in [34], they do yield a non-zero *bias*($\tau$) within our framework. The magnitude of this bias is

small, implying that other perturbations would be necessary to explain an observed, appreciable bias. To relax the neutrality assumption, we simulate recent directional selection. We find that when the selection coefficient is large enough ($4Ns \geq 1$), selection indeed yields biased polygenic scores. Though this selection-induced bias is several orders of magnitude larger than that induced by asymmetry in the detection thresholds, it is still small relative to the variance explained by segregating genetic variants. Additionally, weak selection only induces small deviations from neutral theoretical expectations for the other statistics, suggesting that our neutral theory may still accurately capture accuracy declines in the presence of weak directional selection. Altogether, our theoretical results suggest that allelic turnover may make large contributions to out-of-sample reductions in accuracy, even under neutrality.

## Model and metrics

Our modeling framework readily encompasses two demographic scenarios. In the first, the focal individual is sampled from the same population in which the GWA study was performed, but at a previous point in time $\tau$ (Fig 1A). We specify $\tau$ in coalescent time units: An ancient sampling time of $\tau$ corresponds to $2N \cdot \tau$ generations in the past, with $2N$ as the diploid population size. When $\tau = 0$, the focal individual is an independent sample from the GWA study population. In the second scenario (Fig 1B), the focal individual is sampled at $\tau'$ from a population that diverged from the GWA study population at $\tau_{\text{split}}$ (in coalescent time units) in the past. However, we show in Section 1 in S1 Text that scenario (A) is equivalent to scenario (B) if the ancient sampling time $\tau$ is equal to $2\tau_{\text{split}} - \tau'$. Therefore, we proceed according to the first scenario, while emphasizing that our conclusions readily translate to the second.

We summarize the full model in Fig 1C and detail its constituent parts in the proceeding subsections. Briefly, the genotype of the ancient individual is sampled conditional on the population allele frequencies at $\tau$. The ancient individual's phenotype is then sampled conditional on its genotype. Population allele frequencies for all loci that potentially affect the trait evolve until present day, at which point the GWA study is conducted. In particular, the effect sizes included in the polygenic score model are estimated from the genotypes and phenotypes of $n$ contemporary individuals. Finally, the ancient polygenic score is computed from the ancient individual's genotype and the polygenic score model derived from the results of a contemporary GWA study.

### Sampling the genotype of a time-indexed individual

We assume that each site is at most bi-allelic, with possible alleles $A_1$ and $A_2$. We denote the genotype of an individual sampled at some time $t$ (in coalescent units) as $X_{i\ell}(t)$, where $i$ indexes the individual, and $\ell$ the locus. For the ancient individual(s), $t = \tau$; for the participants in the GWA study, $t = 0$. For mathematical convenience, we use a symmetric genotype encoding, that is $X_{i\ell}(t) \in \{-1, 0, 1\}$, corresponding to genotypes $A_1 A_1$, $A_1 A_2$, and $A_2 A_2$, respectively. Conditional on the population allele frequency of allele $A_2$ at $t$, $Z_\ell(t)$, the distribution of $X_{i\ell}(t)$ is given by the Hardy-Weinberg sampling probabilities: $\mathbb{P}\{X_{i\ell}(t) = -1 | Z_\ell(t) = z\} = (1 - z)^2$, $\mathbb{P}\{X_{i\ell}(t) = 0 | Z_\ell(t) = z\} = 2z(1 - z)$, and $\mathbb{P}\{X_{i\ell}(t) = 1 | Z_\ell(t) = z\} = z^2$.

### Modeling the true phenotype

The genetic basis of a polygenic trait, $Y$, is determined by a set $\mathcal{L}$, consisting of $L$ distinct genetic loci ($|\mathcal{L}| = L$), each with a true per-locus additive effect $\beta_\ell \in \mathbb{R}$ (for $\ell = 1, 2, \ldots, L$). We further assume that the $L$ loci contribute linearly to the trait, such that the true phenotype of

the $i$-th individual sampled at $t$ is specified by the commonly used additive genetic model [37],

$$Y_i(t) = C + \sum_{\ell=1}^{L} X_{i\ell}(t)\beta_\ell + \epsilon_i(t), \tag{1}$$

where $C$ is a constant; $\beta_\ell$ is the true additive effect of locus $\ell$; and $\epsilon_i(t) \sim \mathcal{N}(0, \sigma_e^2)$ is a normally distributed random variable that incorporates variance in the phenotype due to the environment. The summation in Eq 1 is often referred to as an individual's *genetic value* [25]. A locus $\ell$ contributes $\pm\beta_\ell$ to the genetic value (and phenotype) of an individual who is homozygous at $\ell$, and zero to that of a heterozygous individual. $C$ is thus the phenotype of an hypothetical all heterozygous individual. Without loss of generality, we set $C = 0$. In addition, we assume, without loss of generality, that all $\beta_\ell \geq 0$ such that locus $\ell$ contributes $-\beta_\ell$ to the genetic values of $A_1A_1$ individuals and $+\beta_\ell$ to the genetic values of $A_2A_2$ individuals.

A fixed locus, $Z_\ell(t) \in \{0, 1\}$, will affect the mean phenotype of the population at $t$ by $\pm\beta_\ell$ but will not contribute to phenotypic variation. We illustrate this fact by conditioning on the allele frequencies of all loci in $\mathcal{L}$ at $t$, $\mathbf{Z}(t) \in [0, 1]^L$. Assuming linkage equilibrium between loci as well as independence between the environmental and genetic effects, we have,

$$\mathbb{V}[Y_i(t)|\mathbf{Z}(t)] = 2\sum_{\ell=1}^{L} \beta_\ell^2 Z_\ell(t)(1 - Z_\ell(t)) + \sigma_e^2. \tag{2}$$

The summation in Eq 2 is the additive genetic variance at $t$, $V_A(t)$. For a segregating site, the summand is proportional to $Z_\ell(t)(1 - Z_\ell(t))$, with $0 < Z_\ell(t)(1 - Z_\ell(t)) < 1$. For a fixed site, the summand is zero and the site does not contribute to the additive genetic variance $V_A(t)$. An important feature of our model is that some of the $L$ loci may not exhibit genetic variation in the population at a given time. More concretely, the set of loci with non-zero estimated effects on the polygenic score, $\mathcal{S}$, may only be a small subset of $\mathcal{L}$. Thus, we assume that $\mathcal{L}$ is a superset of $\mathcal{S}$.

## Constructing a model for the polygenic score

As our aim is to isolate the effects of allelic turnover on the statistical properties of polygenic scores, we make the additional assumption that the genotyped sites are the causal sites. (We have already assumed that all loci are in linkage equilibrium.) Akin to [38], we employ a simple threshold model for the effect estimates. For a GWA study consisting of $n$ individuals (and $2n$ chromosomes),

$$\hat{\beta}_\ell := \begin{cases} \beta_\ell & \text{if } D_\ell \in (d_{\ell 1}, 2n - d_{\ell 2}), \\ 0 & \text{else,} \end{cases} \tag{3}$$

where $D_\ell$ is the allele count of the trait-increasing allele $A_2$ at the $\ell$-th site in the GWA study sample; and $d_{\ell 1}$ and $d_{\ell 2}$ are the site-specific detection thresholds. In this simplified model, the true effect is estimated perfectly for all sites with allele counts within the intervals ($d_{\ell 1}$, $2n - d_{\ell 2}$) for $\ell \in \mathcal{L}$. In Section 4 in S1 Text, we relate Eq 3 to two alternative estimation procedures: maximum likelihood estimation (MLE) and the best linear unbiased predictor (BLUP).

We allow the two thresholds to differ in order to encompass scenarios in which power is an asymmetric function of the sample allele frequencies, e.g., there is more power to detect low frequency ($D_\ell < n$) versus high frequency ($D_\ell > n$) trait-increasing alleles. Such situations may arise with polygenic disease inheritance and imbalanced case and control sample sizes [34]. In most cases, however, we will consider symmetric detection thresholds, with $d_{\ell 1} = d_{\ell 2} = d_\ell$. The

threshold $d_\ell$ depends on on the phenotypic variance, genome-wide significance threshold, true per-locus effect $\beta_\ell$, and GWA study size $n$. In Section 2 in S1 Text, we give an explicit form for this dependency for a continuous focal trait and equal detection thresholds. Varying $d_\ell$ while keeping the GWA sample size fixed is equivalent to varying the true per-locus effect $\beta_\ell$. Varying the GWA study size $n$ while keeping $\beta_\ell$ and the other parameters fixed is akin to varying the GWA study's power to detect loci of a particular effect size. In **Analytical Results**, we do both.

The threshold model arises in the large GWA study size $n$ limit for the model of $\hat{\beta}_\ell$ provided in Equation 5 in S1 Text. Namely, as long as $D_\ell$ is not too small, the variance of $\hat{\beta}_\ell$ goes to zero as $n$ grows. Thus, the threshold model in Eq 3 will necessarily underestimate the true variance of $\hat{\beta}$ (Section 4 in S1 Text). Still, this model captures the dependency of $\hat{\beta}_\ell$ on the GWA study sample size $n$ and the true per-locus effect $\beta_\ell$, while facilitating our analytical treatment.

In order to compare the polygenic score with an individual's true phenotype, we need to account for all sites in the mutational target $\mathcal{L}$, not just those in $\mathcal{S}$, the set of sites with non-zero effect estimates in the polygenic score. As $\hat{\beta}_\ell = 0$ for any site in $\mathcal{L}$ but not $\mathcal{S}$, we express the polygenic score as a function of all loci in $\mathcal{L}$. The ancient polygenic score of individual $i$ sampled $\tau$ generations in the past is then given by,

$$\hat{Y}_i(\tau) := \hat{C} + \sum_{\ell=1}^{L} X_{i\ell}(\tau)\hat{\beta}_\ell, \tag{4}$$

where $\hat{C}$ is the average phenotype of the GWA sample after subtracting the estimated genetic effects at all loci,

$$\hat{C} := \bar{Y} - \sum_{\ell=1}^{L} \hat{\beta}_\ell \bar{X}_\ell, \tag{5}$$

with $\bar{Y} = \frac{1}{n}\sum_{j=1}^{n} Y_j$ and $\bar{X}_\ell = \frac{1}{n}\sum_{j=1}^{n} X_{j\ell}$ as the mean phenotype and genotype at locus $\ell$ in the GWA study sample, respectively. Here, and in the remainder of our study, we omit time-indexing for random variables associated with the GWA study at $t = 0$. By design, the estimated intercept $\hat{C}$ absorbs the effects of all loci which were not detected as significant in the GWA study, i.e., those sites for which $\hat{\beta}_\ell = 0$. Its presence in the polygenic score of Eq 4 is necessitated by the fact that, to facilitate our analytical treatment, we did not center nor scale the genotypes and phenotypes in the GWA study. Importantly, all of our results are independent of this choice (Section 5 in S1 Text). Henceforth, unless otherwise noted, we refer to Eq 4 as the *polygenic score* and to the summation in Eq 4 as the *genetic prediction*.

## Modeling population genetic dynamics

Population genetic processes govern the correlations between allele frequencies at distinct points in time. We model this correlation using the Wright-Fisher diffusion with recurrent mutation. As we assumed all loci were in linkage equilibrium, their allele frequencies evolve forward in time independently, subject to genetic drift and mutation. At each site, alleles mutate from $A_1 \to A_2$ with rate $\mu$, and from $A_2 \to A_1$ with rate $\nu$. While our results readily generalize to arbitrary $\mu$ and $\nu$, we restrict ourselves to equal mutation rates, $\mu = \nu$.

We further assume that the population is at equilibrium. In this setting, the marginal allele frequencies are beta-distributed, with shape and scale parameters specified by the population-scaled mutation rate; we denote the latter quantity by $a$, with $a = 4N\mu = 4N\nu$.

The relative magnitudes of mutation and genetic drift determine which force dominates an allele frequency trajectory. For example, as $a$ approaches 0, the effects of mutation on the frequencies of segregating mutations become negligble and genetic drift dominates. In this low mutation regime ($a \ll 1$, or equivalently $\mu \ll \frac{1}{2N}$), the recurrent mutation model approximates the infinite sites model, while still retaining the features that make it attractive for our analytical treatment. In particular, the stationary allele frequency distribution is a well-defined probability distribution under the recurrent mutation model, but not under the infinite sites model. We concern ourselves almost exclusively with the low mutation regime.

## Quantifying *out-of-sample* prediction errors

To quantify how well the polygenic score approximates the true phenotype of an individual sampled uniformly at random from the population at time $\tau$ before the present, we use several statistics:

**Bias.** We define the bias as the expectation of the difference between the polygenic score and true phenotype,

$$bias(\tau) := \mathbb{E}[\hat{Y}(\tau) - Y(\tau)], \qquad (6)$$

where, here and elsewhere, we omit the subscript when there is only one sample. The expectation in Eq 6 is with respect to the entire random process, encompassing the underlying population genetic dynamics, estimation of the per-locus effects in the GWA study, and computation of the ancient polygenic score (illustrated in Fig 1C).

**Mean-squared error (*mse*).** We define the *mse* as the expectation of the squared prediction error,

$$mse(\tau) := \mathbb{E}[(\hat{Y}(\tau) - Y(\tau))^2]. \qquad (7)$$

As in Eq 6, the expectation in Eq 7 is with respect to all sources of randomness in the model. The variance of the prediction error equals the difference of the *mse* and the square of the *bias*, and thus it is fully characterized by these two metrics.

**Expected estimated additive genetic variance ($\hat{V}_A$).** The estimated additive genetic variance is an estimate of the amount of phenotypic variance in the ancient population explained by additive genetic effects alone. We use $\hat{V}_A(\tau)$ to represent the expectation of this quantity,

$$\hat{V}_A(\tau) := \sum_{\ell=1}^{L} \hat{V}_{A\ell}(\tau) = 2\sum_{\ell=1}^{L} \mathbb{E}[\hat{\beta}_\ell^2 \hat{Z}_\ell(\tau)(1 - \hat{Z}_\ell(\tau))], \qquad (8)$$

where $\hat{Z}_\ell(\tau)$ is an estimate of the ancient population allele frequency computed from a sample of $n_a$ individuals sampled at $\tau$. The expected true additive genetic variance, $\mathbb{E}[V_A]$, can be found by taking the expectation of the summation in Eq 2.

**Polygenic score accuracy ($\rho^2$).** Practitioners often compute the sample correlation coefficient $r^2$ to measure the accuracy of a predictor in a sample. Here, our sample is $n_a$ ancient individuals sampled at time $\tau$, thus,

$$r^2(\tau) := \frac{Cov[\hat{\boldsymbol{Y}}(\tau), \boldsymbol{Y}(\tau)]^2}{Var[\hat{\boldsymbol{Y}}(\tau)] Var[\boldsymbol{Y}(\tau)]}, \qquad (9)$$

where $Cov[\cdot, \cdot]$ and $Var[\cdot]$ are the sample covariance and variance operators, respectively, and $\hat{\boldsymbol{Y}}(\tau), \boldsymbol{Y}(\tau) \in \mathbb{R}^{n_a}$ are the $n_a$-dimensional vectors of polygenic scores and phenotypes of the

ancient individuals, respectively. Ideally, we would compute the expectation of this quantity—but, this is challenging due to the common difficulty of computing an expectation of a ratio of random variables. Thus, we approximate the expectation of $r^2(\tau)$ as the ratio of expectations,

$$\mathbb{E}[r^2(\tau)] \approx \frac{\mathbb{E}[Cov[\hat{\boldsymbol{Y}}(\tau), \boldsymbol{Y}(\tau)]]^2}{\mathbb{E}[Var[\hat{\boldsymbol{Y}}(\tau)]]\mathbb{E}[Var[\boldsymbol{Y}(\tau)]]} =: \rho^2(\tau), \tag{10}$$

where, as above, the covariance and variances are taken with respect to the sample of $n_a$ ancient individuals, while the expectation is over all sources of randomness in Fig 1C (see Section 7.4 in S1 Text for more details). We present simulations in the section *Polygenic score accuracy* of **Analytical Results** showing that $\rho^2(\tau)$ is a good approximation for the expectation of $r^2(\tau)$ in the parameter regimes of interest.

## Analytical Results

By how much does the prediction accuracy of a polygenic score decrease as the time between sampling the ancient individual and conducting the GWA study increases? To answer this question, we consider a trait potentially influenced by $L$ genetic loci, each with true effect $\beta_\ell \geq 0$, $\ell = 1, \ldots, L$. The forward evolution of sites underlying this trait is modulated by a per site, per generation mutation rate, $\mu$, and a population scaled rate of $a = 4N\mu$. The diploid population of size $2N$ chromosomes is assumed to be at equilibrium. The parameters dictating the GWA study are the sample size $n$ and the detection thresholds specified by $\boldsymbol{d}_1, \boldsymbol{d}_2 \in \{1, \ldots, n\}^L$. The metrics are indexed by the ancient sampling time $\tau$ in coalescent time-units. An ancient sampling time of $\tau$ corresponds to $2N \cdot \tau$ generations in the past. We omit the time index for variables associated with the GWA study, which occurs at present day ($t = 0$). (We show in Section 11 in S1 Text, that the metrics can also be expressed as a function of divergence or $F_{ST}$ between the ancient and contemporary populations).

Each subsection is structured as follows: We first derive a general expression for the statistic that does not depend on how we model the population genetic dynamics nor the GWA study. Second, we derive an analytical expression for the statistic under the population genetic assumptions and the GWA study threshold model described in **Model and metrics**.

### Bias

We can rewrite the sampling time-dependent bias defined in Eq 6 as,

$$bias(\tau) = \sum_{\ell=1}^{L} bias_\ell(\tau) = \sum_{\ell=1}^{L} \mathbb{E}[(\bar{X}_\ell - X_\ell(\tau))(\beta_\ell - \hat{\beta}_\ell)], \tag{11}$$

where $bias_\ell(\tau)$ is the contribution of locus $\ell$ to $bias(\tau)$. From Eq 11, we see that $bias_\ell(\tau) \approx 0$ when either or both of $\hat{\beta}_\ell \approx \beta_\ell$ and $\bar{X}_\ell \approx X_\ell(\tau)$ are true. Thus, $bias_\ell(\tau)$ is minimal when (i) effect estimates are accurate, and (ii) the allele frequencies have not changed substantially in the interval $[\tau, 0]$.

Under the assumption of equal mutation rates and detection thresholds ($d_{\ell 1} = d_{\ell 2}$), $bias_\ell(\tau)$ = 0 for $\tau \geq 0$ for a reason distinct from those stated above. Trait-increasing alleles at high frequencies ($D_\ell > n$) and low frequencies ($D_\ell < n$) are detected as significant ($\hat{\beta}_\ell \neq 0$) with equal probability. An equivalent assumption is that power is not affected by whether the most prevalent allele is trait-increasing or decreasing. Subsequent evolution of the allele frequencies preserves this symmetry and $bias(\tau)$ remains equal to zero for all $\tau$. It follows that in the absence of additional perturbing forces, an estimate of the mean polygenic score from a sample of $n_a$

ancient individuals will also be unbiased, and therefore will on average accurately reflect the lack of change in the mean phenotype.

However, if we introduce asymmetry in the detection thresholds ($d_{\ell 1} \neq d_{\ell 2}$), $bias(\tau)$ is non-zero for all $\tau$ (Section 7.1 in S1 Text). Using the spectral representation of the transition density of the Wright-Fisher diffusion (*tdf*), we derive the per-locus contribution to the bias, $bias_\ell(\tau)$ (Section 7.1 in S1 Text). For a small population-scaled mutation rate $a$ and a large GWA study size $n$, we approximate this expression (given in Equation 45 in S1 Text) as,

$$bias_\ell(\tau) \approx (e^{-a\tau} - 1)(P^{(d_{\ell 1})} - P^{(d_{\ell 2})}),\tag{12}$$

where,

$$P^{(d_{\ell i})} = \sum_{i=0}^{d_{\ell i}-1} \binom{2n}{i} \frac{B(a+i, a+2n-i)}{B(a,a)}\tag{13}$$

is the probability that the allele count of site $\ell$ is less than $d_{\ell i}$, i.e., $D_\ell < d_{\ell i}$ for $i = 1, 2$; and, $B(\cdot, \cdot)$ is the beta function. Thus, the magnitude of $bias_\ell(\tau)$ is approximately proportional to the difference in the probability of detecting high ($D_\ell > n$) versus low ($D_\ell < n$) frequency alleles, and increases exponentially with $\tau$. With a large GWA study size $n$ and a small mutation rate $a$, this difference is small relative to the square root of the additive genetic variance—the ratio of these two quantities is smaller than $\mathcal{O}(a)$ (Fig S1a in S1 Text). This is due to the fact that when the mutation rate is small, most alleles are close to fixation or fixed. The stationary population allele frequency density $\kappa(z) \propto z^{a-1}(1-z)^{a-1}$ behaves like $z^{-1}(1-z)^{-1}$ for small $a$. Varying $d_{\ell i}$ then has relatively little impact on $P^{(d_{\ell i})}$, constraining the difference between the one-sided detection probabilities (Fig S1b in S1 Text).

## Mean-squared error

The sampling time-dependent mean-squared error $mse(\tau)$ can be expressed as,

$$
\begin{aligned}
mse(\tau) \quad &= \sum_{\ell=1}^{L} mse_\ell(\tau) + \left(\frac{n-1}{n}\right)\sigma_e^2 \\
&= \sum_{\ell=1}^{L} \mathbb{E}\left[(X_\ell(\tau) - \bar{X}_\ell)^2(\hat{\beta}_\ell - \beta_\ell)^2\right] + \left(\frac{n-1}{n}\right)\sigma_e^2,
\end{aligned}
\tag{14}
$$

where $\sigma_e^2$ is the variance in the phenotype due to the environment (Section 7.2 in S1 Text). Note the similarity of the left term in Eq 14 to the form of $bias(\tau)$ given in Eq 11—similar heuristics apply. Under the threshold model specified in Eq 3, sites at moderate frequencies in the GWA study sample, $D_\ell \in [d_\ell, 2n - d_\ell]$, will not contribute to $mse(\tau)$ since $\hat{\beta}_\ell = \beta_\ell$. Only sites with frequencies outside this interval (including sites invariant in the GWA study sample) will contribute, and their contributions will be proportional to the squared difference between $X_\ell(\tau)$ and $\bar{X}_\ell$. In practice, moderate frequency loci will also contribute to $mse(\tau)$ due to errors in the estimation of the effect estimates and any difference between the ancient genotypes and the average genotypes in the GWA study sample at these sites (Section 4 in S1 Text).

We use the spectral representation of the *tdf* (Section 6 in S1 Text) to derive an analytical expression for $mse_\ell(\tau)$, the per-locus contribution to the *mse* (Section 7.2 in S1 Text). From this expression, Equation 50 in S1 Text, we derive a linear approximation for the initial per-locus increase in this statistic, $\Delta mse_\ell(\tau)$. With a symmetric detection threshold ($d_{\ell 1} = d_{\ell 2} = d_\ell$)

we have,

$$\Delta mse_\ell(\tau) := mse_\ell(\tau) - mse_\ell(0) \approx 2\beta_\ell^2 a P^{(d_\ell)}\tau, \tag{15}$$

where $mse_\ell(0)$ is the contribution of site $\ell$ to $mse(\tau)$ for $\tau = 0$ (Equation 76 in S1 Text); and $2P^{(d_\ell)}$, defined in Eq 13, is the probability that the allele count of site $\ell$ is outside the detection interval such that $\hat{\beta}_\ell = 0$. Both $mse_\ell(0)$ and $P^{(d_\ell)}$ depend on the mutation rate $a$, the GWA study size $n$, and the detection threshold $d_\ell$.

$\Delta mse_\ell(\tau)$ reflects the time-dependent contributions of sites *not* detected in the GWA study. To see this, we condition on the effect estimate $\hat{\beta}_\ell$, $mse_\ell(\tau) = \beta_\ell^2 \mathbb{E}[(X_\ell(\tau) - \bar{X}_\ell)^2 | \hat{\beta}_\ell = 0] \cdot 2P^{(d_\ell)} + 0 \cdot (1 - 2P^{(d_\ell)})$. Thus, Eq 15 implies that $\frac{d\mathbb{E}[(X_\ell(\tau) - \bar{X}_\ell)^2 | \hat{\beta}_\ell = 0]}{d\tau} \approx a$ for small $\tau$, and consequently, the combined effects of drift and mutation on $mse_\ell(\tau)$ are captured in the product of the mutation rate and sampling time $a\tau$.

In addition, Eq 15 suggests that the rate at which $mse_\ell(\tau)$ increases will be shared across parameter regimes when $aP^{(d_\ell)}$ is similar (Fig S4a in S1 Text). To illustrate this, we use our analytic formula (given in Equation 50 in S1 Text) to compute $mse_\ell(\tau)$ for several low mutation rates, $a \in \{10^{-4}, 10^{-3}, 10^{-2}\}$, and three GWA study sizes, $n \in \{10^4, 10^5, 10^6\}$ (Fig 2A). These mutation rates and sample sizes span the range of parameter values appropriate for human data. We depict our results in two ways: (i) we plot the change in $mse_\ell(\tau)$, and (ii) we plot $mse_\ell(\tau)$ normalized by the expected additive genetic variance contributed by a single site. At stationarity the expected additive genetic variance is constant and equal to,

$$\mathbb{E}[V_{A\ell}] = \mathbb{E}[2\beta_\ell^2 Z_\ell(1 - Z_\ell)] = \beta_\ell^2(a/(2a + 1))$$

for a scaled-mutation rate $a$. The plot of the former, Fig 2A, exhibits the functional relationship revealed by Eq 15, while the latter, Fig 2B, approximates the noise-to-signal ratio. In Section 9 in S1 Text, we demonstrate that Eq 15 is a good approximation to $mse(\tau)$ for $\tau \leq 0.2$, particularly when the GWA study size $n$ is large (in particular, see Fig S5 in S1 Text).

To find the GWA study size specific detection thresholds used in Fig 2A and 2B, we solve Equation 11 in S1 Text for a given effect size $\beta$, phenotypic variance $V_p$, and significance threshold $\alpha$, while varying the GWA study sample size. For $\beta^2 = 0.01$, $V_p = 1$, and $\alpha = 10^{-8}$, the detection thresholds are $d = 4142, 3340, 3290$ in order of increasing sample size, which corresponds to sample allele frequencies of approximately 0.2, 0.02, amd 0.002, respectively. Thus, for a given effect size, larger sample sizes will lead to the detection of alleles at more extreme allele frequencies, while smaller samples will restrict detection to alleles at more intermediate frequencies. Due to non-identifiability, the parameter choices are fairly arbitrary.

We find that for small mutation rates, the cumulative change in the *mse*, $\Delta mse_\ell(\tau)$, is mostly insensitive to differences in the GWA study sample size (Fig 2A and 2B). The approximation in Eq 15 helps to explain this result. The rate of increase is approximately proportional to $2aP^{(d_\ell)}\tau$. For small mutation rates ($a \ll 1$) and an arbitrary detection threshold $d_\ell$, the probability of *not* detecting a locus as significantly associated with the trait is roughly $2aP^{(d_\ell)} \approx 1$ for all sufficiently large $n$ (Fig S1b in S1 Text). In this regime, increasing the GWA study sample size only yields small increases in the probability of detecting a locus as significant. Thus, for small mutation rates, the product of this quantity with the mutation rate is $2aP^{(d_\ell)} \approx a$, and indeed, we observe a cumulative increase in $mse_\ell(\tau)$ that is $\mathcal{O}(a)$ for $\tau = 1$ (Fig 2A). We note that increasing the GWA study sample size does enable detection of loci with smaller effects.

The result in Fig 2A, however, hides the fact that a small absolute increase in $mse(\tau)$ may correspond to a substantial increase in the noise-to-signal ratio. Indeed, for $a = 10^{-3}$ (blue lines throughout), $mse_\ell(\tau)$ ultimately exceeds the expected additive genetic variance $\mathbb{E}[V_{A\ell}]$ for

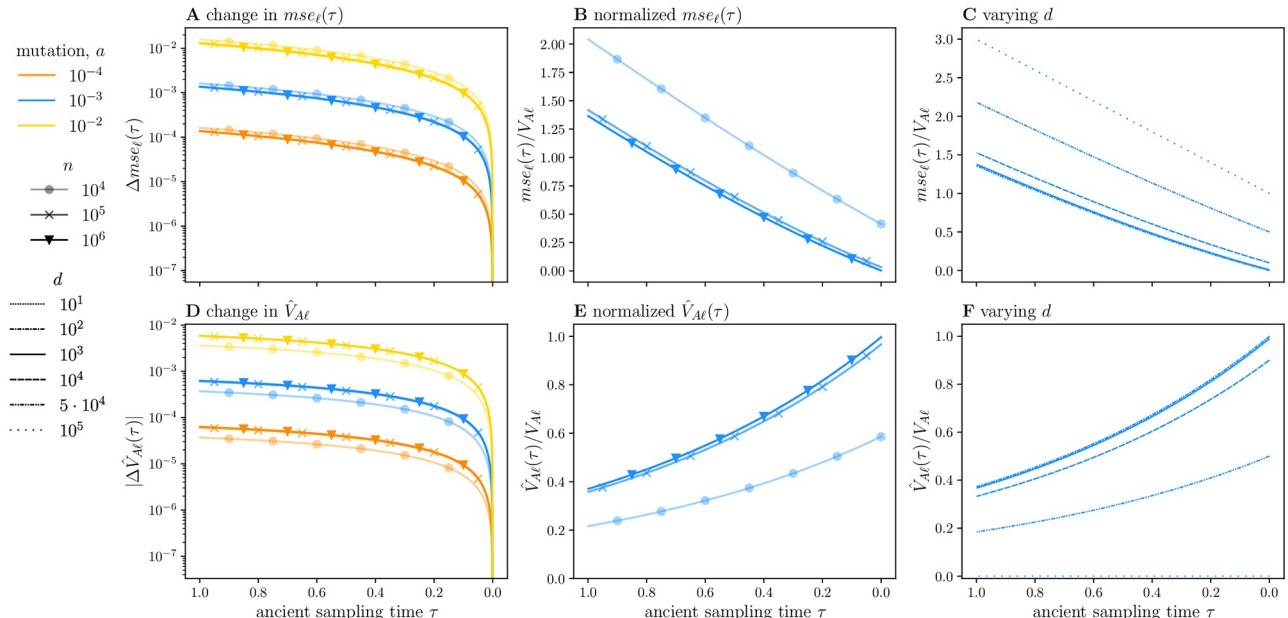

**Fig 2. Per locus contributions to the mean-squared error and estimated additive genetic variance across sample sizes, mutation rates, and detection thresholds.** In (A), we plot the per-locus increase in $mse$, $\Delta mse_\ell(\tau)$, normalized by $\beta^2$, for three mutation rates $a = 10^{-4}, 10^{-3}, 10^{-2}$ by color, and for the three sample sizes, $n = 10^4, 10^5, 10^6$ by shape, respectively. For a squared effect size of $\beta^2 = 0.01$, each sample size, in part, specifies a value of $d_\ell$, with $d = 4142, 3340, 3290$, or sample allele frequencies of approximately 0.2, 0.02, and 0.002, in order of increasing sample size. In (B-C), we restrict ourselves to $a = 10^{-3}$ as the lines for different mutation rates would otherwise largely coincide. In (B), we plot $mse_\ell(\tau)$ normalized by the expected additive genetic variance at stationarity, $\mathbb{E}[V_{A\ell}] = \beta^2 a/(2a + 1)$. In (C), we fix $n = 10^4$ and vary the detection threshold over several orders of magnitude, $d \in \{10, \ldots, 10^5\}$, plotting $mse_\ell(\tau)$ normalized by $\mathbb{E}[V_{A\ell}]$. In (D-F), we repeat (A-C), but for the statistic $\hat{V}_{A\ell}(\tau)$, with the following exception: Because $\hat{V}_{A\ell}(\tau)$ decreases with $\tau$, we plot the absolute value of its difference from $\hat{V}_{A\ell}(0)$ in (A). For all plots the ancient sampling time $\tau \in [1, 0]$, which corresponds to a time span of $2N$ generations.

all GWA study sample sizes (Fig 2B). By $\tau = 0.2$, a sampling time characteristic of ancient humans, $mse_\ell(\tau)$ due to allelic turnover is approximately 20% of the additive genetic variance $\mathbb{E}[V_{A\ell}]$. For sufficiently large $\tau$, $mse_\ell(\tau)$ is at least the same order of magnitude as the expected additive genetic variance. In addition, while $mse_\ell(\tau)$ increases at approximately the same rate irrespective of study size, its initial value $mse_\ell(0)$ is sample size dependent (Fig 2B and see Fig S4b and S4e in S1 Text for a larger parameter space). Yet, for a given value of $d_\ell$, reductions in $mse_\ell(0)$ mediated by sample size diminish once $n$ is large enough (Fig S4b and S4e in S1 Text).

Further, Fig 2A obscures the fact that different mutation rates may yield similar noise-to-signal ratios. As discussed, for small $a$, $mse_\ell(\tau)$ increases with $\tau$ at a rate that is $\mathcal{O}(a)$. For small $a$, the additive genetic variance is likewise $\mathcal{O}(a)$, yielding a relative increase that is mostly insensitive to the mutation rate. Normalized $mse_\ell(0)$ is also similar across small mutation rates (Fig S4b and S4e in S1 Text), rendering relative $mse_\ell(\tau)$ mostly insensitive to $a$. We thus omitted the other two mutation rates from Fig 2B.

Lastly, we fix the GWA study sample size at $n = 10^5$ and vary the detection threshold $d$ (Fig 2C). Varying $d$ while keeping $n$ fixed is analogous to varying the true per-locus effect size $\beta$, or keeping $\beta$ fixed while varying the significance threshold $\alpha$. The minimum threshold is $d = 10$, whereas $d = n = 10^5$ maximizes $mse_\ell(\tau)$ since $\hat{\beta}_\ell$ would equal zero for all $\ell$. Consistent with our analysis above, for small $a$, (i) $mse_\ell(0)$ depends critically on $d$, while (ii) $mse_\ell(\tau)$'s approximately linear growth rate is largely insensitive to $d$. Furthermore, by our previous arguments, relative $mse_\ell(\tau)$ is similar across small mutation rates, and they are also omitted in Fig 2C. For

independent and identically distributed (*iid*) loci and $\sigma_e^2 = 0$, the per-locus $mse_\ell(\tau)$ values presented in Fig 2B and 2C are equal to the corresponding trait-wide statistics $mse(\tau)$.

## Additive genetic variance

The per-locus contribution to the expected estimated additive genetic variance $\hat{V}_A(\tau)$ is,

$$\hat{V}_{A\ell}(\tau) = 2\mathbb{E}\left[\hat{\beta}_\ell^2 \hat{Z}_\ell(\tau)(1 - \hat{Z}_\ell(\tau))\right] = 2\left(\frac{2n_a - 1}{2n_a}\right)\mathbb{E}\left[\hat{\beta}_\ell^2 Z_\ell(\tau)(1 - Z_\ell(\tau))\right], \tag{16}$$

where $\hat{Z}(\tau) = \frac{1}{2n_a}\sum_{i=1}^{n_a}(X_i(\tau) + 1)$ is the estimated allele frequency at $\tau$, computed in a sample of $n_a$ ancient individuals. When $\hat{\beta}_\ell = 0$ or $Z_\ell(\tau) \in \{0, 1\}$, site $\ell$ will not contribute to $\hat{V}_A(\tau)$. Thus, a site $\ell$ has a non-zero contribution to the estimated additive genetic varianceonly when it is segregating at both the present day and $\tau$. This condition is necessary for both $\hat{Z}_\ell(\tau)(1 - \hat{Z}_\ell(\tau)) > 0$ and $\hat{\beta}_\ell \neq 0$ to be true.

As with the two previous statistics, we use the spectral representation of the *tdf* to derive an analytical expression for $\hat{V}_A(\tau)$ under our population genetic assumptions (Section 7.3 in S1 Text). The resulting expression, Equation 54 in S1 Text, indicates that the expected additive genetic variance decays exponentially. We then, to first order in the ancient sampling time $\tau$, approximate the initial decrease in the per-locus estimated additive genetic variance $\Delta\hat{V}_{A\ell}(\tau)$,

$$\Delta\hat{V}_{A\ell}(\tau) := \hat{V}_{A\ell}(\tau) - \hat{V}_{A\ell}(0) = -2\left(\frac{2n_a - 1}{2n_a}\right)\beta_\ell^2 a P^{(d_\ell)}\tau, \tag{17}$$

where $\hat{V}_{A\ell}(0)$ is $\hat{V}_{A\ell}(\tau)$ evaluated at $\tau = 0$ (Equation 77 in S1 Text);and $2P^{(d_\ell)}$, defined in Eq 6, is the probability that $\hat{\beta}_\ell = 0$. The factor due to finite sampling, $2n_a/(2n_a - 1)$, is $\approx 1$ when the ancient sample size $n_a$ is large. Thus, apart from sign, $\Delta\hat{V}_{A\ell}(\tau)$ is equal to $\Delta mse_\ell(\tau)$ of Eq 15. Therefore, for small $\tau$, $\hat{V}_A(\tau)$ decreases at approximately the same rate as $mse(\tau)$ increases. This result further suggests that for $a \ll 1$ and a large GWA study size $n$, $\hat{V}_{A\ell}(\tau)/\mathbb{E}[V_{A\ell}] \approx 1 - mse_\ell(\tau)/\mathbb{E}[V_{A\ell}]$ for small $\tau$ (Fig 2C and 2F). Although, this relationship trivially breaks down for large $\tau$ as $mse_\ell(\tau)$ is not bounded by one.

To compare $\hat{V}_{A\ell}(\tau)$ across mutation rates, we mirror our treatment of $mse_\ell(\tau)$ in the previous section. We plot (i) its increase $\Delta\hat{V}_{A\ell}(\tau)$ (Fig 2D); (ii) $\hat{V}_{A\ell}(\tau)$ normalized by the expectation of the true additive genetic variance at stationarity (Fig 2E); and (iii) normalized $\hat{V}_{A\ell}(\tau)$, varying the detection threshold for a fixed GWA study sample size (Fig 2F). Akin to $mse_\ell(\tau)$, normalized $\hat{V}_{A\ell}(\tau)$ is very similar across small mutation rates. And, while the GWA study size $n$ and the detection threshold $d$ influence the initial estimated additive genetic variance $\hat{V}_{A\ell}(0)$, its rate of change is mostly insensitive to the two GWA study parameters.

As $\hat{V}_A(\tau)$ largely recapitulates our results for $mse(\tau)$ with opposing sign, we focus on their differences. Indeed, they have different functional forms and behave differently for modest or large $\tau$ (see Equations 50 and 54 in S1 Text, respectively). Conceptually, this discrepancy is not unexpected: In the previous section, we showed that a site only contributes to $mse(\tau)$ if its allele count falls outside the detection interval and $\hat{\beta}_\ell = 0$. Thus, $mse(\tau)$ increases with $\tau$ due to alleles shifting from intermediate frequencies in the ancient population to frequencies *outside* of the detection region in the contemporary population. For the expected estimated additive genetic variance $\hat{V}_A(\tau)$, the converse is true: The slope represents the decline in $\hat{V}_A(\tau)$ due to alleles changing from frequencies near or at fixation in the ancient population to frequencies *within* the detection interval in the contemporary population. While our results reveal similar

functional behavior for these two quantities (with opposing signs) that applies for small $\tau$, we caution that statements about $\hat{V}_A(\tau)$ do not immediately translate to statements about $mse(\tau)$, particularly for $\tau \gtrapprox 0.2$.

## Polygenic score accuracy

While our framework, in principle, encompasses a trait with varying effect sizes, we will first assume that all sites are *iid* with true effect size $\beta$. Our approximation to the expectation of the sample correlation coefficient simplifies to,

$$\rho^2(\tau) = \frac{L\beta\mathbb{E}[\hat{\beta}(X(\tau) - \bar{X}(\tau))^2]}{L\beta^2\mathbb{E}[(X(\tau) - \bar{X}(\tau))^2] + \sigma_e^2} = \frac{\mathbb{E}[\hat{\beta}Z_\ell(\tau)(1 - Z_\ell(\tau))]/\beta}{\mathbb{E}[Z_\ell(\tau)(1 - Z_\ell(\tau))] + \sigma_{e'}^2}, \quad (18)$$

where the compound parameter $\sigma_{e'}^2 = \sigma_e^2/L\beta^2$ is the environmental variance normalized by the product of the number of loci in the mutational target $L$ and the squared per-locus effect size $\beta$ (Section 7.4 in S1 Text). By comparing Eq 18 with Eq 16, we can see that $\rho^2(\tau)$ is closely related to the estimated additive genetic variance. Thus, like $\hat{V}_A(\tau)$, $\rho^2(\tau)$ will decrease with $\tau$ due to loci having changed from frequencies close to zero or one in the ancient population to intermediate frequencies in the contemporary population. However, unlike $\hat{V}_A(\tau)$, $\rho^2(\tau)$ does not depend on the ancient sample size. Therefore, to relate the two statistics, we multiply by the inverse of the ancient sample size dependent factor implicit in $\hat{V}_A(\tau)$,

$$\rho^2(\tau) = \left(\frac{2n_a}{2n_a - 1}\right) \frac{\hat{V}_{A\ell}(\tau)/\beta^2}{\mathbb{E}[V_{A\ell}(\tau)]/\beta^2 + \sigma_{e'}^2}. \quad (19)$$

For $\sigma_e^2 = 0$, barring the sample size factor, Eq 19 is equal to $\hat{V}_A(\tau)$ normalized by the expected additive genetic variance. By extension, this quantity approximates the expected sample correlation coefficient $r^2(\tau)$. By invoking our additional population genetic and GWA study assumptions, we arrive at an approximation for the decrease in polygenic score accuracy,

$$\Delta\rho^2(\tau) := \rho^2(\tau) - \rho^2(0) \approx -\frac{2aP^{(d_\ell)}\tau}{\frac{a}{2a+1} + \sigma_{e'}^2}. \quad (20)$$

Now, to relate our theory to empirical and simulation studies, we compute $\rho^2(\tau)$ for a given narrow-sense heritability $h^2$ and mutation rate $a$ pair. We define $h^2$ for a trait with a mutational target of $L$ loci of equal effects $\beta$,

$$h^2 := \frac{\mathbb{E}[V_A]}{\mathbb{E}[V_A] + \sigma_e^2} = \frac{a/(2a+1)}{a/(2a+1) + \sigma_{e'}^2},$$

where the equality follows from our population genetic assumptions. Together with $a$, $h^2$ fully specifies the compound parameter $\sigma_{e'}^2$ with,

$$\sigma_{e'}^2 = \left(\frac{a}{2a+1}\right)\left(\frac{1 - h^2}{h^2}\right).$$

We plot our analytical expressions for both accuracy (Fig 3A) and relative accuracy (Fig 3B), defined as the ratio of $\rho^2(\tau)$ to $\rho^2(0)$ for $\tau \in [1, 0]$ spanning $2N$ generations. For humans, this time span corresponds to approximately 500,000 years in the past, encompassing the "Out-of-Africa" migration event estimated to have occurred 50,000–100,000 years ago [39]. As

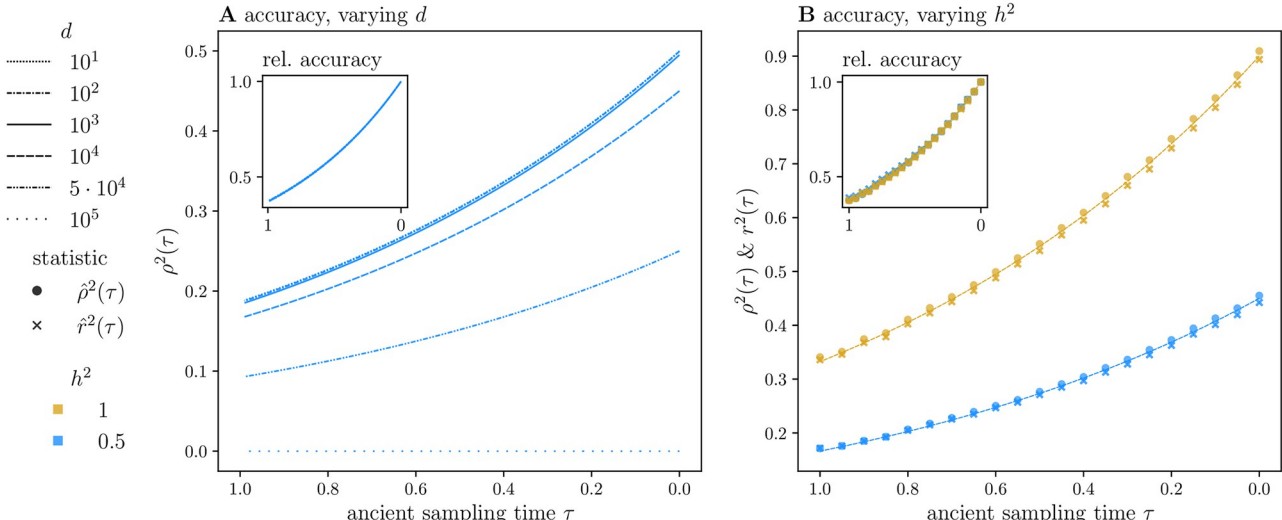

**Fig 3. Polygenic score accuracy.** We plot our theoretical results for both absolute (A, main) and relative accuracy $\rho^2(\tau)$ (A, inset) for ancient sampling times $\tau \in [1, 0]$ (or a time span of $2N$ generations) with a mutation rate of $a = 10^{-3}$. The GWA study size is shared in all plots, with $n = 10^5$. In (A), we vary the detection threshold over the range of possible values, $d_\ell \in \{10, \ldots 10^5\}$. In (B), we compare our theoretical expectations with simulated estimates of the approximate sample correlation coefficient $\rho^2(\tau)$ (circles) and the statistic itself $r^2(\tau)$ (crosses) for a threshold of $d = 10^4$ (a minimum sample allele frequency of 0.05), and two values of heritability, $h^2 = 0.5, 1$ (in blue and gold, respectively). The ancient sample size is $n_a = 100$. In the inset of (B), we normalize the estimates by their initial (estimated) values. Theoretical expressions for $\rho^2(\tau)$ are also plotted in (B). Each simulated point is the average of $K = 5000$ simulations of $L = 5000$ *iid* loci.

with the preceding statistics, when $\tau = 0$, $\rho^2(\tau)$ approximates the accuracy of the polygenic score *within* the GWA study population. Relative accuracy then directly measures reductions in accuracy relative to the GWA study population. We set $h^2 = 0.5$ and $a = 10^{-3}$, and fix the GWA study sample size at $n = 10^5$. We then compute $\rho^2(\tau)$, varying the detection threshold over several orders of magnitude (Fig 3A). (See Fig S6 in S1 Text for accuracy as a function of the fixation index, or $F_{ST}$.) Our results for $\rho^2(\tau)$ necessarily recapitulate those of $\hat{V}_A(\tau)$: While increasing the detection threshold $d$ reduces accuracy substantially, it does not have a large impact on relative accuracy for $n = 10^5$ (Fig 3A). Indeed, for small mutation rates, relative accuracy is insensitive to the mutation rate and threshold, and is well approximated by $e^{-\tau}$ (Equation 68 in S1 Text). Thus, its derivative is also exponential. Absolute accuracy $\rho^2(\tau)$ likewise decays exponentially, but its derivative is scaled by a quantity that reflects features of the GWA study and the phenotypic variance. For a small mutation rate $a \ll 1$, its derivative is approximately $2P^{(d)}(a/(a + \sigma_{e'}^2))e^{-\tau}$, which, in turn, is approximately $2P^{(d)}h^2e^{-\tau}$ (Equation 67 in S1 Text). The latter expression suggests that the probability of not detecting a significant association $P^{(d)}$ and trait heritability $h^2$ are the key determinants of prediction accuracy. Importantly, $\rho^2(\tau)$ declines considerably over the interval $\tau \in [1, 0]$ irrespective of the detection threshold $d$.

In addition, we glean from Eq 18 that while heritability affects the magnitude of $\rho^2(\tau)$ through the compound parameter $\sigma_{e'}^2$, it does not influence the relative accuracy, consistent with previous results [16]. Our simulations suggest that this is also true of the sample correlation coefficient, as simulated estimates of $r^2(\tau)$ agree extremely well with our theory for $\rho^2(\tau)$ (Fig 3B). We note that this result is contingent on the fact that the environmental variance $\sigma_e^2$ only enters our simple threshold model in the specification of the threshold $d$ (Equation 11 in S1 Text), and does not contribute directly to the variance of the polygenic score (Section 7.4 in S1 Text). Therefore, we expect this result to hold only for large GWA study sample sizes for

which the threshold model is a good approximation to the distribution of $\hat{\beta}$. While the finding that relative accuracy is insensitive to the GWA study parameters relies on the assumption that all loci are *iid* and share a causal effect $\beta$, we provide preliminary theoretical evidence that our results will hold when $\beta$ varies across loci (see Equation 69 in S1 Text and ensuing comments).

## Simulation results for recent directional selection

We use simulations to explore if and how the statistics under study deviate from their neutral expectations in the presence of recent directional selection. Each copy of the $A_2$ allele at the $\ell$-th site confers a fitness advantage of $+s_\ell$, and so the fitness ratio of the three possible genotypes $A_1A_1$:$A_1A_2$:$A_2A_2$ is $1$:$(1 + s_\ell)$:$(1 + 2s_\ell)$. In our simulations, the population evolves neutrally until the onset of selection at $N$ generations (or $\tau_s = 0.5$ in coalescent time units) before present. Thereafter, the population evolves according to discrete Wright-Fisher dynamics with selection.

In the presence of selection, the allele frequency distributionis no longer symmetric; rather, it is skewed toward the beneficial allele. The severity of the skew depends on the selection coefficient and mutation rate, as well as the amount of time that selection has been acting. As we restrict $s_\ell$ to positive values, designatingthe $A_2$ or + allele as beneficial, the allele frequency distribution will be skewed toward one. If we instead designated the $A_1$ allele as the beneficial allele, the allele frequency distribution would be skewed toward zero. The former models "positive" selection whereas the latter models "negative" selection. Because $bias(\tau)$ is proportional to $\beta$, its sign will be sensitive to this choice, but its magnitude will be unaltered. The other statistics will not be affected as long as the detection thresholds are symmetric. Therefore, our results are general up to the sign of $bias(\tau)$.

We conduct simulations over a range of selection coefficients, $\sigma = 4Ns \in \{0, 0.1, 1, 10\}$, for a mutation rate of $a = 10^{-3}$. Under directional selection, $\sigma$ is proportional to the locus effect size $\beta$; mutations with larger effect sizes will be more likely to establish and achieve appreciable frequencies [40]. In addition, we plot results for two different detection thresholds, $d \in \{10^3, 10^4\}$, in a GWA study sample of size $n = 10^4$. More details on the simulation procedures are provided in Section 3 in S1 Text.

When $\sigma \geq 1$, the polygenic score is biased towards positive values for $\tau > 0$ for both detection thresholds (Fig 4A). In other words, with directional selection acting to increase the trait value, $\hat{Y}(\tau)$ tends to overestimate $Y(\tau)$. The magnitude of $bias_\ell(\tau)$ depends critically on the strength of selection relative to mutation: We observe a larger bias for $\sigma = 10$ relative to $\sigma = 1$, and likewise the bias is larger for $\sigma = 1$ relative to $\sigma = 0.1$. In fact, the smaller selection coefficient $\sigma = 0.1$ is not distinguishable from neutral expectations. For $0 \leq \tau < \tau_s$, $bias_\ell(\tau)$ increases at an accelerating rate; for $\tau \geq \tau_s$, $bias(\tau)$ appears constant in this parameter regime.

A higher detection threshold decreases the detection probability. Thus, we expect that the magnitude of $bias_\ell(\tau)$ will increase with the detection threshold. Indeed, $bias_\ell(\tau)$ is larger and increases more quickly for the larger detection threshold $d = 10^4$ compared to $d = 10^3$ (Fig 4A). Further, our simulations suggest that the detection threshold coupled with the time of the onset of selection govern the magnitude of the bias for $\tau > \tau_s$. For some large $\tau$, $bias_\ell(\tau)$ will reach an equilibrium value that depends approximately on the asymmetry of the detection thresholds at the present day, which in turn, depends on both the timing and strength of selection (Section 10 in S1 Text).

The underlying allele frequency dynamics provide some insight into these patterns. Before the onset of selection, the allele frequency distribution is stationary and symmetric around 0.5. After the onset of selection, trait-increasing alleles tend to increase in frequency, skewing the distribution toward one. Thus, alleles *not* detected in the GWA study will tend be at higher

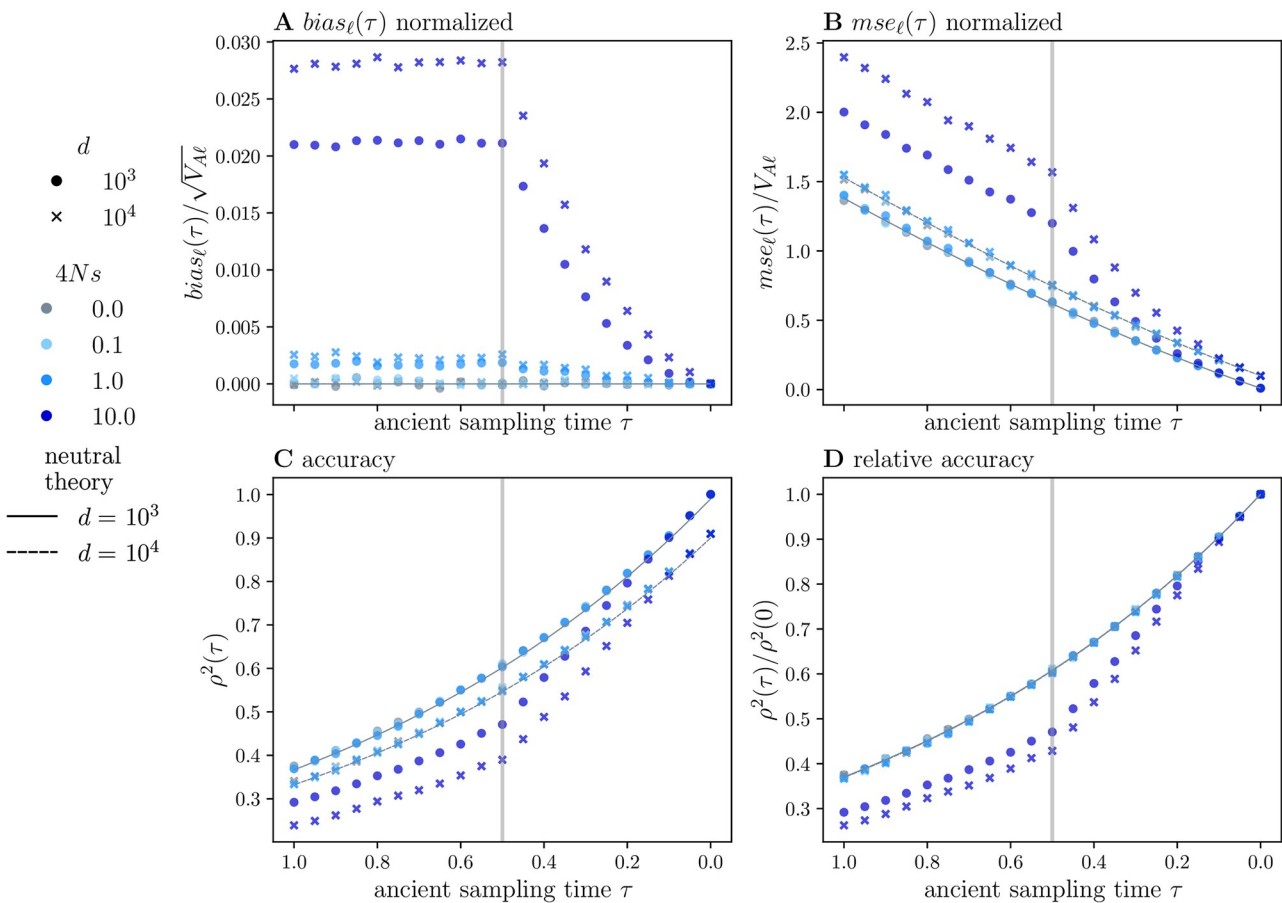

**Fig 4. Ancient polygenic scores in the presence of genic selection.** We conduct $K = 5000$ simulations, each with a mutational target of $L = 5000$ loci, in a population of size $2N = 2 \cdot 10^3$, with a population-scaled mutation rate, $a = 10^{-3}$. We consider four selection coefficients, $\sigma = 4Ns \in \{0, 0.1, 1, 10\}$ (indicated by color). The GWA study sample size is $2n = 2 \cdot 10^5$, with $d$ equal to either $10^3$ or $10^4$. In (A-D), we plot the various simulated statistics along with their neutral expectations (solid or dashed black lines). The vertical gray lines indicate the onset of selection at $\tau_s = 0.5$ which corresponds to $N = 1000$ generations. The ancient sample times are $\tau \in [1, 0]$, corresponding to a time span of $2N = 2000$ generations. We computed, but did not plot, 95% confidence intervals for $bias(\tau)$, $mse(\tau)$, and $r^2(\tau)$, as they largely overlapped with the symbols. We note that the oscillations observed in (A) and (B) are not statistically significant.

versus lower frequencies at $t = 0$, yielding $\mathbb{E}[\bar{X}_\ell | \hat{\beta}_\ell = 0] > 0$ for $\sigma > 0$. For large $\tau$, the allele frequencies of sites not detected in the GWA study, i.e., with $\hat{\beta}_\ell = 0$, may have been substantially different in the ancient population. Each one of these sites will make a contribution to $bias(\tau)$ that is proportional to $\beta_\ell \mathbb{E}[(\bar{X}_\ell - X_\ell(\tau)) | \hat{\beta}_\ell = 0]$ (Eq 11). Looking backward in time, the shift in the allele frequency distribution ensures that the conditional expectation of $X_\ell(\tau)$ is smaller than that of $\bar{X}_\ell$, yielding a positive $bias_\ell(\tau)$ for $\tau > 0$. Notably, the magnitude of $bias_\ell(\tau)$ induced by selection is several orders of magnitude larger than that induced by asymmetry in the detection threshold alone (Fig S1a in S1 Text).

The effects of selection on $mse_\ell(\tau)$ are qualitatively consistent with those on $bias_\ell(\tau)$ (Fig 4B). Although, here, the only selection coefficient which induces significant deviations from neutral expectations is $\sigma = 10$. And, $mse(\tau)$ is larger for $d = 10^4$ compared to $d = 10^3$. As with $bias(\tau)$, for $0 \le \tau < \tau_s$, $mse_\ell(\tau)$ increases at an accelerating rate; before $\tau_s$ ($\tau \ge \tau_s$), $mse_\ell(\tau)$ appears to increase linearly. Values of $\sigma < 10$ do not induce noticeable deviations from neutrality for the correlation coefficient $\rho^2(\tau)$ either (Fig 4C). However, strong selection ($\sigma = 10$)

does lead to substantially larger reductions in accuracy relative to our neutral expectations. In addition, for $\sigma = 10$, relative accuracy is sensitive to the detection threshold, with accuracy decreasing faster for the larger detection threshold (Fig 4D).

## Discussion

In this work, we devised a theoretical framework to quantify the effect of allelic turnover on the error and accuracy of out-of-sample polygenic scores. Unlike previous theoretical approaches [16, 27], we averaged over the evolutionary process governing trait evolution, the GWA study from which a polygenic score model is constructed, and the ancient individual's genotype and phenotype. In doing so, we found explicit expressions for several commonly used metrics that depend on the focal individual's sampling time, as well as the parameters governing the population genetic dynamics and power to detect trait-associated loci in the GWA study. Mathematical properties of the recurrent mutation model at stationarity enabled us to compute analytical expressions for the metrics of interest under neutrality, and approximations thereof.

Our analytical expressions suggest that allelic turnover alone may be responsible for large reductions in accuracy: For small mutation rates, $\rho^2(\tau)$ (and $r^2(\tau)$) decreases substantially within short time-spans, by about 20 percent in $0.2N$ generations (corresponding to approximately 120,000 years in humans). In addition, increasing the detection threshold yielded lower polygenic score accuracy, as a locus was less likely to have a non-zero effect. These results are broadly consistent with a concurrent study by Yair and Coop [41], in which the authors used simulations to assess cross-population prediction accuracy, defined as the ratio of the variance of and individual's polygenic score to that of their genetic value, under neutrality and in the presence of stabilizing selection. When Yair and Coop restricted the polygenic score to the top one percent of SNPs, roughly analogous to altering the detection threshold, they similarly found that the accuracy declined in the focal population.

Yet, while the detection threshold influenced the magnitude of the polygenic score accuracy, relative accuracy was insensitive to this parameter. In other words, under neutrality, relative accuracy is insensitive to the magnitude of the per-locus effect and only depends on the underlying allele frequency distribution. In addition, relative accuracy was independent of the size of the mutational target when the constituent loci were *iid*. Our theory suggests that these results will hold for arbitrary distributions of the true effect $\beta$. Consideration of several effect size distributions in a parameter regime consistent with the UK Biobank further supports this conjecture (Section 8 in S1 Text). Although more work is required to fully substantiate this claim.

Selection, however, induces a dependency between an allele's effect and its frequency, and may thereby render relative accuracy sensitive to the detection threshold. Our simulations provide preliminary evidence in support of this claim. For a small mutation rate of $a = 4N\mu = 10^{-3}$ and a large per-locus selection coefficient $\sigma = 4Ns = 10$, relative accuracy was lower for the larger detection threshold of $d = 10^4$ compared to $d = 10^3$. Yet, the difference between detection thresholds was small relative to that induced by selection, and was negligible for smaller selection coefficients. Indeed, smaller selection coefficients ($\sigma \leq 1$) did not yield appreciable deviations from our neutral expectations for the *mse*, accuracy, nor relative accuracy. Therefore, excluding strong selection ($\sigma > 1$), our neutral expectations for these statistics appear to be good approximations to their true values. Our theoretical results under neutrality thus may prove an accurate description of temporally-resolved polygenic scores when polygenic adaptation is achieved by concurrent small frequency changes at numerous small effect loci—a plausible scenario [28, 35]. In addition, the simple patterns revealed by our simulations suggest

that it may be possible to derive (approximate) analytic expressions for the given metrics in the presence of strong selection, when loci exhibit selective sweep-like behavior.

It is unclear whether our neutral expectations will hold in the context of more sophisticated polygenic trait modeling. In our simulation study, as in our theoretical work, we focus on dynamics at a single locus. Thus, our results are most relevant to scenarios in which single locus dynamics can be decoupled from the evolution of the mean phenotype and the genetic background [40]. Namely, the effect of an individual locus must be small relative to the mean phenotype [38, 40]. Future work will assess polygenic score accuracy under more sophisticated models of polygenic adaptation (e.g., [38, 42]).

Of the two bias-inducing processes explored, detection threshold asymmetry and directional selection, the latter induced much larger deviations from our neutral expectation for the bias, i.e., under neutrality $bias(\tau) = 0$ for all ancient sampling times $\tau$. In the presence of detection asymmetry, $bias(\tau)$ is approximately proportional to the difference between the one-sided detection probabilities, which in turn is constrained by the shape of the allele frequency distribution. Under neutrality, and for small mutation rates, most alleles are at very low frequencies or fixed, such that changing the detection threshold minimally influences the one-sided detection probabilities. Selection, however, perturbs the underlying allele frequency density. At equilibrium, this density is proportional to $e^{\sigma z}z^{-1}(1-z)^{-1}$ for small $a$, where $\sigma = 4Ns$. Depending on $\sigma$, the one-sided detection probabilities may differ markedly, yielding larger values of $bias(\tau)$. We thus suspect that detection asymmetry has the potential to further exacerbate any bias induced by selection. These results are interesting in light of those of Chan et al. 2014 [34], who demonstrated that polygenic disease inheritance under the liability threshold model induced differences in the power to detect protective versus susceptible alleles. In Chan et al., this effect was further increased by imbalances in the case and control sample sizes in the GWA study. Additional work is needed to incorporate these features of case-control studies into our modeling framework.

The effects of selection on the bias have implications for assessments of mean differences between ancient polygenic scores from distinct time points. In particular, our results suggest that sufficiently strong positive directional selection will lead to overestimation of the difference between the polygenic scores of ancient individuals sampled before and after the onset of selection. Likewise, in the presence of negative selection, the polygenic score will underestimate this difference. At the same time, as discussed above, estimation error increases (as measured by $mse(\tau)$) and accuracy (as measured by $\rho^2(\tau)$) decreases as the ancient sampling time increases.

Our results clarify relationships between various commonly used metrics of prediction error and accuracy. For example, we demonstrated an approximate functional relationship between the mean-squared error $mse(\tau)$ and the expected additive genetic variance $\hat{V}_A(\tau)$ that applies for small ancient sampling times and mutation rates. This shared initial rate emerged despite fundamental differences between these statistics: $mse(\tau)$ measures error due to variants near or at fixation in the contemporary sample, which were segregating at intermediate frequencies in the ancient sample. In contrast, $\hat{V}_A(\tau)$ measures error due to variants segregating in the contemporary sample, which were near or at fixation in the ancient sample. This conceptual result does not rely on any of our population genetic or GWA modeling assumptions, and perhaps could be exploited to learn about the genetic architecture of quantitative traits from multi-population data. In addition, we showed formally that polygenic score accuracy $\rho^2(\tau)$, an approximation to the expectation of the sample correlation coefficient $r^2(\tau)$, is proportional to the ratio of $\hat{V}_A(\tau)$ to the total phenotypic variance. We believe that these relations, and their evolutionary and GWA study dependent forms, may facilitate the development of

novel, more principled statistical procedures for the analysis of out-of-sample polygenic scores.

At the same time, the simplifying assumptions underlying our results indicate that significant challenges remain. For one, our model does not incorporate the complex demographic processes, such as admixture and population size changes, inherent in human history. This implies that an ancient sampling time of $t$ years in the past likely does not correspond to a sampling time of $\tau = t/2N$ in our model, where $2N$ is the contemporary population size. Indeed, allelic turnover cannot explain all of the reductions in accuracy observed in out-of-sample predictions in humans. For example, our neutral theory predicts an approximately fifty percent reduction in accuracy when $F_{ST}$ between the focal and GWA study populations is comparable to African-European divergence ($F_{ST} \approx 0.1$). This more severely overestimates the prediction accuracy of height in a sample of individuals with African ancestry compared to the Wang et al. predictions, which take into account both LD and allele frequency changes (Section 12 in S1 Text). Thus, to achieve the same accuracy reductions observed in both simulated, e.g., [15, 16] and empirical, e.g., [14, 16, 43], studies of cross-population polygenic scores for contemporary humans, allelic turnover under neutrality would require population divergence times that far exceed their estimated values (Fig S7 in S1 Text).

Differences in LD between contemporary human populations may largely explain this discrepancy as most trait-associated loci are likely to be tagging rather than causal sites [12, 16]. As with geographically distinct populations, if LD between the genotyped and causal sites differed in the ancient population, then polygenic score accuracy would suffer [1]. We did not model this effect and assumed that the genotyped site was the causal site. This assumption may be justified when ancient sampling or population divergence times are recent, as high marker density in the GWA study may mitigate accuracy losses due to LD decay, but more theoretical work is required to substantiate this claim. While our framework can readily incorporate LD, it is difficult to obtain analytical results when the genotyped marker is *not* the causal site. In lieu of theoretical results, large-scale simulations in simple population genetic scenarios may provide insight into the relative contributions of LD—which depends on the allele frequencies of the tagging and causal sites—and allelic turnover to declines in polygenic score accuracy.

Furthermore, our assumption of linkage equilibrium between loci roughly equates to assuming that each LD block contains only a single causal site. Thus, our results will be most applicable to traits with relatively sparse genetic architectures for which the distance between any two causal loci is large compared to the scale of LD. In contrast, when the trait architecture is dense, a large number of variants have non-zero effect on the trait. Causal sites in close proximity are necessarily linked, and our assumption of linkage equilibrium would be violated. In addition, under a dense trait architecture, the "prune and threshold" polygenic score described herein may achieve lower accuracy than a best linear unbiased predictor (BLUP) that allows all segregating loci to have non-zero effects. In Section 4 in S1 Text, we speculate on the accuracy of BLUP in the context of our modeling framework when the trait has a dense architecture.

In addition, we assumed that per-locus causal effects were shared by the ancient and contemporary samples. Differences in causal effects across contemporary populations, perhaps due to changes in the environment, epistasis, or gene-by-environment interactions, likely contribute to accuracy reductions [8, 12]. Indeed, Cox et al. [18] found that trends in the polygenic scores of temporally disparate ancient samples did not always recapitulate those of the true phenotype. We conjecture that fluctuations in the per-locus effects would increase $mse(\tau)$ and decrease accuracy, but not profoundly alter our conclusions. Perhaps, if the fluctuations were asymmetric, e.g., effect sizes tended to increase in time, then $bias(\tau)$ may be non-zero under neutrality. Population stratification in the GWA study population may also lead to biased ancient polygenic scores, as has been observed in cross-population predictions in humans [9,

10]. Lastly, technical challenges inherent to the extraction and sequencing of ancient DNA often result in noisy estimates of the ancient genotypes. This additional source of randomness is likely to reduce accuracy and increase $mse(\tau)$, but otherwise should not substantially alter our conclusions.

## Supporting information

**S1 Text. Extended model, methods, and results.** This supplementary text contains detailed derivations and additional analyses.
(PDF)

## Acknowledgments

We thank members of the Berg, Novembre, and Steinrücken labs, and the Cummings fourth floor for helpful discussions throughout the development of this project. In addition, we thank Jennifer Blanc, Adam Fine, Evan Koch, Zachary Miller, and John Novembre for comments on earlier (or very early) versions of this manuscript. We also give a special thanks to Carlos A. Serván and Micol Tresoldi for numerous insightful discussions over the course of this project.

## Author Contributions

**Conceptualization:** Maryn O. Carlson, Daniel P. Rice, Matthias Steinrücken.

**Data curation:** Maryn O. Carlson.

**Formal analysis:** Maryn O. Carlson, Daniel P. Rice, Matthias Steinrücken.

**Funding acquisition:** Matthias Steinrücken.

**Investigation:** Maryn O. Carlson, Matthias Steinrücken.

**Methodology:** Maryn O. Carlson, Daniel P. Rice, Jeremy J. Berg, Matthias Steinrücken.

**Software:** Maryn O. Carlson.

**Supervision:** Matthias Steinrücken.

**Validation:** Maryn O. Carlson.

**Visualization:** Maryn O. Carlson.

**Writing – original draft:** Maryn O. Carlson.

**Writing – review & editing:** Maryn O. Carlson, Daniel P. Rice, Jeremy J. Berg, Matthias Steinrücken.

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
