## [Decision Letter · Decision Letter 0]

24 Nov 2021

Dear Dr Steinrücken,

Thank you very much for submitting your Research Article entitled 'Polygenic score accuracy in ancient samples: quantifying the effects of allelic turnover' to PLOS Genetics.

The manuscript was fully evaluated at the editorial level and by independent peer reviewers. The reviewers appreciated the attention to an important topic but identified some concerns that we ask you address in a revised manuscript.

We therefore ask you to modify the manuscript according to the review recommendations. Your revisions should address the specific points made by each reviewer.

[LINK]

Yours sincerely,

Kirk E Lohmueller

Guest Editor

PLOS Genetics

Bret Payseur

Section Editor: Evolution

PLOS Genetics

Thank you for submitting this work to PLOS Genetics. It has been evaluated by 3 reviewers. All generally liked the manuscript. The reviewers provide a number of comments to improve the manuscript. None stood out as being more critical than others. I think addressing these comments, even those that were presented as suggestions/not required by the reviewers, in your revision will improve the readability and applicability of your work.

Reviewer's Responses to Questions

**Comments to the Authors:**

Reviewer #1: This manuscript by Carlson and colleagues considers the effect of "allelic turnover" on polygenic scores fit to ancient samples. "Allelic turnover," as the authors define it, is relevant to the problem of polygenic score portability in part because causal variants that explain trait variation in an ancient sample may experience frequency changes that make them undetectable in modern GWAS. The authors use a diffusion approach to study several summaries of polygenic score performance. I have not checked all the equations, but the work appears to be well done. I have a few comments, which I list not as requirements for publication (which I leave up to the editor), but rather as friendly reactions and suggestions.

1) The model here assumes that the ancient sample is from a population directly ancestral to a contemporary population, when in fact many ancient samples may not fit this description straightforwardly. (See e.g. Schraiber 2018, PMID: 29167200). I think it is totally fair for the authors to model things the way they do, as it captures a lot of the intuition and allows them just to think about one forward diffusion. However, the paper might benefit from a clearer discussion of this potential caveat.

2) Along similar lines to (1), I was struck by how much of the framing is in terms of ancient samples when a lot of the intuitions developed are relevant to pairs of contemporary populations. The model is more straightforwardly related to ancient populations (though see (1)), but I think it'd be interesting for the authors to give more space to thinking about the relevance for contemporary populations. There is a paragraph on this in the discussion, but only a hint of a quantitative comparison is given, and as far as I can tell mainly for empirical results. It would be useful to include a comparison with the modeling approach of Wang, Guo et al (ref 16).

3) arXiv:1909.00892 has a nice discussion of the relevance of allelic turnover to polygenic score comparisons as proxies for phenotype comparisons. This is captured in the various summaries studied, but I would have liked more emphasis on distinguishing within-population accuracy vs. getting changes in the mean over time correct in the discussion.

4) A new preprint by Yair & Coop is relevant for this study and takes a complementary modeling approach https://doi.org/10.1101/2021.09.10.459833 . (I know what this is starting to look like, and I swear I'm not Graham Coop.)

5) I found the explanations of the results in the main text to be, in some places, a little wordy and hard to follow. I think it is worthwhile to communicate the parts of the intuition that have to do with allele frequencies changing into or out of the GWAS-detectable range, but I got lost in some of the verbal explanations. I also thought the summary in the intro was in too much depth for not having seen the model yet. The repeated structure of the results subsections, though helpful for comprehension, also get to feel redundant. I would urge the authors to cut down the main text, lean more on their figures, and even possibly make use of tables to summarize some of the main results (e.g. in the intro).

6) I appreciated that the authors brought in literature on genetic prediction outside humans.

7) The final paragraph is perhaps missing a mention of environmental differences and GxE.

Minor comments

a) line 139, I assume that N is a diploid population size because time units are scaled in 2N generations, but please specify.

b) line 167, is it meant that this quantity is often called a "genetic value" specifically in this paper? I have not seen it much in other places. If it's meant that it's a general term, please provide a citation.

c) p. 5, I might have missed it, but I don't see it explicitly stated here that eq. 2 assumes linkage equilibrium as well. (It shows up in the beginning of 2.3 but should probably be here first.)

Reviewer #2: In this manuscript, Carlson et al. provide a masterful assessment of polygenic score accuracy decreases due to allelic turnover, in the context of ancient populations that are ancestral to the GWAS panel on which a phenotype was measured. They quantify this decrease as a function of various parameters of interest, including the temporal distance between the ancient and present-day population, the power to detect significantly-associated loci, and trait heritability. They also produce simulations to determine how much weak selection on trait-associated variants biases the quantities of interest.

The manuscript is well written and it is clear that the authors have done a great deal of work to reach their conclusions, and evaluate various quantities of interest for geneticists to assess how reliable a score for an ancient individual would be. I only have very minimal suggestions for changes or additions to the manuscript, that I don't deem crucial for the manuscript to be accepted for publication:

To provide a clearer picture for researchers choosing to do (or not do) polygenic prediction on ancient samples, it would be nice to work with values from a concrete phenotype and GWAS study: given the size and heritability estimates from the latest UK Biobank GWAS or Biobank Japan on a particular phenotype (e.g. height), how much would one mis-estimate this phenotype in a population that is X years away from UKBB or BBJ, in the past, solely due to allelic turnover?

Rarely are polygenic scores in the ancient DNA literature computed for populations that are directly ancestral to the present-day GWAS population, as there has been substantial population turnover and admixture in human populations during the Holocene and Pleistocene. It might be worth emphasizing this more strong in the Discussion so that the reader is aware that a population sampled X years in the past (in the same location as a present-day population) does not necessarily correspond to your scenario of an ancestral population that is separated by X years from the present-day GWAS populations.

In this vein, the manuscript would benefit from moving some of the results from Supplementary Text S1.1 to the main text (at a minimum the figure and the final conclusion), as those would more generally applicable to the larger set of researchers not necessarily working with directly ancestral populations, or working with diverged populations from the GWAS one, outside the ancient genomic literature.

Following up on this, could the authors also express the turnover-caused decreased in accuracy in terms of Fst values, rather than the ancient sampling time in coalescent units? This would make it a lot easier for an empirical researcher to assess how much of a decrease in accuracy due to allelic turnover they would expect for their ancient (or out-of-space) polygenic predictions, given that Fst values between present-day and ancient populations are readily obtainable (assuming stationarity and that allelic turnover is the only contributor to the decrease in accuracy).

I would be curious to see how accuracy decreases with increasing the number of ancient individuals used in the computation of polygenic score, e.g. how much more / less problematic is it to compute the mean polygenic score for a population of ancient hunter-gatherers vs. computing a polygenic score individually for each ancient hunter-gatherer in the population. How is the variance for these different estimates affected by the sampling time?

Beyond recent directional selection, could the authors provide some intuition as to how persistent negative selection would affect their results? i.e. How would these be affected when the magnitude of the effect size estimates determines the probability that the individual variants will be removed from the population?

Reviewer #3: See attached

**Have all data underlying the figures and results presented in the manuscript been provided?**

Reviewer #1: Yes

Reviewer #2: Yes

Reviewer #3: Yes

PLOS authors have the option to publish the peer review history of their article (what does this mean?). If published, this will include your full peer review and any attached files.

Reviewer #1: No

Reviewer #2: **Yes: **Fernando Racimo

Reviewer #3: **Yes: **Luke J. O'Connor

---

## [Decision Letter · Decision Letter 1]

3 Mar 2022

Dear Dr Steinrücken,

Thank you very much for submitting your Research Article entitled 'Polygenic score accuracy in ancient samples: quantifying the effects of allelic turnover' to PLOS Genetics.

The manuscript was fully evaluated at the editorial level and by independent peer reviewers. The reviewers appreciated the attention to an important topic but identified some concerns that we ask you address in a revised manuscript.

We therefore ask you to modify the manuscript according to the review recommendations. Your revisions should address the specific points made by each reviewer.

[LINK]

Yours sincerely,

Kirk E Lohmueller

Guest Editor

PLOS Genetics

Bret Payseur

Section Editor: Evolution

PLOS Genetics

Thank you for submitting your revised manuscript. Two of the three reviewers are satisfied with the changes. Reviewer 3 has some additional comments on the BLUP model in the Supplement. Please provide some qualitative description of what would change under a BLUP-type estimator with no threshold.

Reviewer's Responses to Questions

**Comments to the Authors:**

Reviewer #1: The authors have addressed my first-round comments sufficiently, and I have no further comments. This was a nice paper even before the revision.

Reviewer #2: The authors have addressed all of my concerns. I recommend this paper for acceptance.

Reviewer #3: In their revised manuscript, Carlson et al. have addressed most of my comments from the initial submission, but I was not completely satisfied by their treatment of the BLUP PRS estimator, which I still think would produce qualitatively different results than the threshold-style PRS estimator they primarily analyze.

The approximation the authors use for the PRS is that there is some detection threshold based on the true effect size (and AF) of a variant, and we get a good estimate of the effect size for variants above the threshold, but a zero estimate for variants below it. This approximation is aligned with a threshold-style estimator, where you throw out variants using a threshold for their estimated (rather than true) effect size/significance.

It contrasts with the BLUP estimator, which does not throw out any variants. The new supplementary note seems to be arguing that the BLUP is approximately the same as the estimator under consideration because its weights are well approximated by the OLS effect size estimates. This is true (perhaps even obvious), but the OLS effect size estimates are only used in the threshold approximation *for variants passing the threshold*, and this threshold seems to be critical reason for the qualitative findings throughout the paper.

As I noted before, the difference between these estimators corresponds to the difference between a sparse vs. infinitesimal genetic architecture: under a sparse architecture, a method with some sort of feature selection is called for; under an infinitesimal architecture, when every SNP is causal, the BLUP estimator is optimal.

I do not at all think it would detract from the findings of the paper if they are only applicable to sparse architectures and threshold/variable selection style estimators. We know that real genetic architectures are not infinitesimal (especially for non-psychiatric traits), and in general, models with some thresholding or variable selection perform better than BLUP.

I don’t request new analyses to address this, but I am concerned that the new supplementary note misses the mark, and I recommend at least some qualitative description of what would change under a BLUP-type estimator with no threshold.

**Have all data underlying the figures and results presented in the manuscript been provided?**

Reviewer #1: None

Reviewer #2: Yes

Reviewer #3: Yes

PLOS authors have the option to publish the peer review history of their article (what does this mean?). If published, this will include your full peer review and any attached files.

Reviewer #1: No

Reviewer #2: **Yes: **Fernando Racimo

Reviewer #3: **Yes: **Luke J O'Connor

---

## [Decision Letter · Decision Letter 2]

26 Mar 2022

Dear Dr Steinrücken,

We are pleased to inform you that your manuscript entitled "Polygenic score accuracy in ancient samples: quantifying the effects of allelic turnover" has been editorially accepted for publication in PLOS Genetics. Congratulations!

Yours sincerely,

Kirk E Lohmueller

Guest Editor

PLOS Genetics

Bret Payseur

Section Editor: Evolution

PLOS Genetics

Comments from the reviewers (if applicable):

Reviewer's Responses to Questions

**Comments to the Authors:**

Reviewer #3: The authors have addressed my comments.

**Have all data underlying the figures and results presented in the manuscript been provided?**

Reviewer #3: Yes

PLOS authors have the option to publish the peer review history of their article (what does this mean?). If published, this will include your full peer review and any attached files.

Reviewer #3: **Yes: **Luke J O'Connor

**Data Deposition**

http://datadryad.org/submit?journalID=pgenetics&manu=PGENETICS-D-21-01313R2

**Press Queries**

---

## [Editor Report · Acceptance letter]

26 Apr 2022

PGENETICS-D-21-01313R2 

Polygenic score accuracy in ancient samples: quantifying the effects of allelic turnover 

Dear Dr Steinrücken, 

We are pleased to inform you that your manuscript entitled "Polygenic score accuracy in ancient samples: quantifying the effects of allelic turnover" has been formally accepted for publication in PLOS Genetics! Your manuscript is now with our production department and you will be notified of the publication date in due course.

With kind regards,

Andrea Szabo

PLOS Genetics

On behalf of:
